# Current and Potential Therapeutic Options for Infections Caused by Difficult-to-Treat and Pandrug Resistant Gram-Negative Bacteria in Critically Ill Patients

**DOI:** 10.3390/antibiotics11081009

**Published:** 2022-07-26

**Authors:** Helen Giamarellou, Ilias Karaiskos

**Affiliations:** 1st Department of Internal Medicine-Infectious Diseases, Hygeia General Hospital, 4, Erythrou Stavrou & Kifisias, Marousi, 15123 Athens, Greece; e.giamarellou@hygeia.gr

**Keywords:** pandrug-resistant, *Klebsiella pneumoniae*, *Acinetobacter baumannii*, *Pseudomonas aeruginosa*, salvage treatment, double carbapenem, newer β-lactam-β-lactamase inhibitors, cefiderocol, eravacycline, antimicrobial combinations

## Abstract

Carbapenem resistance in Gram-negative bacteria has come into sight as a serious global threat. Carbapenem-resistant Gram-negative pathogens and their main representatives *Klebsiella pneumoniae*, *Acinetobacter baumannii*, and *Pseudomonas aeruginosa* are ranked in the highest priority category for new treatments. The worrisome phenomenon of the recent years is the presence of difficult-to-treat resistance (DTR) and pandrug-resistant (PDR) Gram-negative bacteria, characterized as non-susceptible to all conventional antimicrobial agents. DTR and PDR Gram-negative infections are linked with high mortality and associated with nosocomial infections, mainly in critically ill and ICU patients. Therapeutic options for infections caused by DTR and PDR Gram-negative organisms are extremely limited and are based on case reports and series. Herein, the current available knowledge regarding treatment of DTR and PDR infections is discussed. A focal point of the review focuses on salvage treatment, synergistic combinations (double and triple combinations), as well as increased exposure regimen adapted to the MIC of the pathogen. The most available data regarding novel antimicrobials, including novel β-lactam-β-lactamase inhibitor combinations, cefiderocol, and eravacycline as potential agents against DTR and PDR Gram-negative strains in critically ill patients are thoroughly presented.

## 1. Introduction

Antimicrobial resistance poses a major threat to human health all over the world. Τhe global burden associated with bacterial antimicrobial resistance in 2019 was an estimated 4.95 million deaths, of which 1.27 million were directly attributable to drug resistance. There is an emphasis on six common pathogens accountable for nosocomial infections: *Escherichia coli*, *Staphylococcus aureus*, *Klebsiella pneumoniae*, *Streptococcus pneumoniae*, *Acinetobacter baumannii*, and *Pseudomonas aeruginosa*, which were responsible for 73% of deaths attributable to antimicrobial resistance in the same report [1]. Additionally, carbapenem resistance in Gram-negative bacteria has come into sight as a serious global threat [2]. The 2017 World Health Organization (WHO) global priority list of pathogens ranks carbapenem-resistant Enterobacteriaceae (CRE), carbapenem-resistant *Pseudomonas aeruginosa*, and carbapenem-resistant *Acinetobacter baumannii* in the highest priority category [3]. More recent attention has focused on evidence of increased likelihood of morbidity and mortality in patients infected by carbapenem-resistant pathogens in comparison to those infected by susceptible pathogens [4,5]. A new terminology has been proposed for the categorization of resistance in Gram-negative pathogens. Multi-drug resistant (MDR) is defined as the acquired nonsusceptibility to at least one agent in three or more categories of antimicrobial agents, and extensively-drug resistant (XDR) is the nonsusceptibility to at least one agent in all but two or fewer categories of antimicrobial agents. Finally, PDR is the nonsusceptibility to all agents in all categories of antimicrobial agents [6]. This statement was proposed by Magiorakos et al. [6] in 2012, when new β-lactam-β-lactamase inhibitors and novel antimicrobial agents were not launched in the market for the treatment of MDR, XDR, and PDR Gram-negative pathogens [7]. Therefore, a new consensus to be established in the era of novel β-lactam-β-lactamase inhibitors is of great matter. However, a new definition of resistance for Gram-negative infections defined as difficult-to-treat resistance (DTR) has recently been proposed as treatment-limiting resistance to all first-line agents, including all β-lactams (carbapenems and β-lactamase inhibitor combinations) and fluoroquinolones [8]. On the other hand, there is a considerable knowledge gap for the treatment of PDR Gram-negative strains, which are linked to extremely high all-cause mortality, ranging from 20 to 71% [9]. Therapeutic options for DTR and PDR *K. pneumoniae*, *A. baumannii*, and *P. aeruginosa* are scarce and based exclusively on few case reports and small case series, initiating salvage treatments counting upon synergistic combinations (in vitro or animal model), increased exposure regimen adapted to the MIC of the pathogen, as well as the introduction of novel antibacterial agents [9].

A narrative review of relevant studies was conducted using the PubMed/MEDLINE, Scopus, and Web of Science databases (from 1970 up to January 2022). The keywords used alone or in combination were pandrug, pandrug-resistant, pan-resistant, epidemiology of PDR, difficult to treat, difficult-to-treat-resistance, salvage treatment, Gram-negative limited options, compassionate use, double carbapenems, ICU patients, critically ill patients, novel β-lactam-β-lactamase inhibitors, cefiderocol, and eravacycline. Information regarding therapy of DTR and PDR Gram-negative infections were included. Full text and abstract screening as well as review articles were searched.

In this review, the latest data regarding the current and potential therapeutic choices for DTR and PDR Gram-negative bacteria are reported and discussed.

## 2. Carbapenem-Resistant *Klebsiella pneumoniae*

### 2.1. Epidemiological Issues

In a detailed review of 125 PDR *K. pneumoniae* strains, the geographical distribution was as follows: (i) Europe (71 strains), Greece being the predominant European country (47 strains), accompanied by Italy, France, and the Netherlands; (ii) America (12 strains); (iii) Asia (41 strains), mostly in India (28 strains). Only one strain was observed in Australia and none from Africa [8]. Regarding all-cause mortality, PDR *K. pneumoniae* strains, despite therapeutic manipulations, were reported as lethal in 31% of bloodstream infections (BSI), 50% in respiratory tract infections (RTIs), 29% in complicated urinary tract infections (cUTIs), 100% in CNS and complicated intra-abdominal infections (cIAI), and 67% in osteomyelitis, with a total fatality rate of 47%. The high mortality rates reported are referred to critically ill patients with high severity scores, with almost 37% of the patients hospitalized in the ICU [9].

#### 2.1.1. Salvage Therapies

Salvage treatments for PDR infections caused by Gram-negative pathogens have been analyzed in a retrospective single-center cohort study, including 65 consecutive eligible patients suffering from infections with a PDR profile hospitalized at the University Hospital of Heraklion, Crete, Greece, between January 2010 and June 2018 [10]. Of the 65 PDR isolates, 31 (48%) were *K. pneumoniae*, followed by *A. baumannii* (43%), and *P. aeruginosa* (9%). All strains were resistant to all available antimicrobial agents; however, the mechanism of resistance was not reported. The majority of the patients were hospitalized in the ICU (79%) with multiple comorbidities, whereas severe sepsis and septic shock at the onset of infection was reported in 14% and 22% of cases, respectively. The most common empirical therapy was colistin-based combination, followed by non-colistin, non-tigecycline combination, and carbapenems plus tigecycline. Empiric therapy was defined arbitrarily as “effective empirical therapy” in cases where antimicrobial treatment administered (although in vitro non-susceptible) before the microbiological documentation of the PDR infection resulted in clinical improvement, without the necessity of treatment modification. The empirical therapy was effective in 50%, 37.5%, and 8% of patients receiving colistin combination, carbapenems-tigecycline, and non-colistin, non-tigecycline combination, respectively (*p* = 0.003). The infection-related in-hospital mortality was 32%. Even though the authors do not distinguish empirical therapeutic results regarding *K. pneumoniae*, *P. aeruginosa*, and *A. baumannii*, the obtained cure rates support the use of colistin and/or tigecycline-based combinations as empirical therapy when an infection due to PDR pathogens is suspected [10]. However, the frequent use of the pre-reported older antibiotics has provoked the emergence of strains with high resistance rates, particularly towards colistin; a fact attributed mainly to overconsumption [11]. In another retrospective study from Greece, amongst 412 monomicrobial BSIs due to *K. pneumoniae*, 115 (27.9%) were due to PDR isolates. The majority of infections were primary BSIs (46.1%), followed by catheter-related BSI (30.4%), cIAI (9.6%), and ventilator-associated pneumonia (VAP) (7.0%). *bla*_KPC_ was the most prevalent carbapenemase gene (85.2%), followed by a co-carriage of *bla*_KPC_ and *bla*_VIM_ (6.1%), *bla*_VIM_ (5.2%), and *bla*_NDM_ (3.1%). Thirty-day mortality was 39.1%. Among all patients, multivariate analysis identified the development of septic shock, Charlson comorbidity index, and BSI other than primary or catheter-related as independent predictors of mortality, while a combination of at least three antimicrobials was identified as an independent predictor of survival for PDR infections caused by *K.pneumoniae* [12].

#### 2.1.2. Double Carbapenem Combinations (DCC)

The rationale of the application of the so-called DCC, i.e., “Double Carbapenem Combination” in case of PDR or XDR *K. pneumoniae* infections, was based on “ertapenem higher affinity with the carbapenemase enzyme, acting as a suicide inhibitor, thus allowing higher levels of the other carbapenems (meropenem or doripenem) to be active in the vicinity of the pathogen” [13]. The first worldwide report was from Greece in 2013 including 3 ICU patients with complicated UTIs [14], to be followed by another study, comprising 27 Greek patients with untreatable infections suffering from cUTIs with secondary bacteremia (four), primary (six) or catheter related BSI (two), hospital acquired pneumonia (HAP) or ventilator associated pneumonia (VAP) (two), and external ventricular drainage infection (one) [15]. PDR strains were isolated in 14 cases, whereas in the remaining 13 cases an XDR profile was identified. Fifteen patients were hospitalized in the ICU and twelve in the medical ward. The median APACHE score was 17 and the median Charslon index was 3, whereas 41% of the cases presented with severe sepsis or septic shock. Patients were treated exclusively with ertapenem (1 g daily, 1-h infusion, to be administered 1-h prior to meropenem dose) and high-dose prolonged infusion meropenem (2 gr, 3-h infusion, every 8-h). MICs against meropenem ranged between 2 and ≥16 mg/L. Clinical and microbiological success was 77.8% and 74.1%, respectively, with an attributable mortality of 11.1%. The results are independent of the height of meropenem MICs. Subsequently, until 2020 ninety patients, after combining ertapenem either with meropenem or doripenem, were published with a successful clinical outcome of 65.5%, and a rather low mortality of 24.2% [15,16,17,18,19,20]. Although the department of hospitalization was not reported in the majority of cases, all patients were critically ill and at least 53 cases were reported to be hospitalized in the ICU [20]. Despite difficulties in evaluation, the beneficiary addition of another antibiotic (mostly colistin) to which the isolated strains of *K. pneumoniae* were resistant in vitro, should also be mentioned [19,20].

#### 2.1.3. The Novel β-Lactamase Inhibitors

In the chapters to follow, the novel β-lactamase inhibitors combination currently in the market (i.e., ceftazidime/avibactam, meropenem-vaborbactam, imipenem-cilastatin-relebactam) and the forthcoming aztreonam-avibactam are presented and discussed, focusing mainly on clinical issues dealing with DTR pathogens in critically ill patients and ICU patients, illustrated in Table 1. Mechanism of action, spectrum of activity, mechanism of resistance, approved indications, and information on DTR and PDR Gram-negative pathogens are depicted in Table 1 [21,22,23,24,25,26,27,28,29,30,31,32,33,34,35,36,37,38,39,40,41,42,43,44,45,46,47,48,49,50,51,52,53,54,55,56,57,58,59,60,61,62,63,64,65,66,67,68,69,70,71]. Although in vitro these agents have demonstrated susceptibility against PDR strains [72], clinical experience is limited to case reports, if any applicable. Nonetheless, these newer agents have the potency for treatment of DTR pathogens; however, more clinical studies focusing on PDR *K.pneumoniae* infections are needed.

### 2.2. Clinical Experience with Diazabicyclooctanes Based β-Lactamase Inhibitors (DBO Inhibitors)

#### 2.2.1. Ceftazidime-Avibactam

Avibactam, a novel non-β-lactam-β-lactamase inhibitor, restores the activity of ceftazidime against the majority of β- lactamases, as outlined in Table 1. In Greece, around 2014–2016, against a collection of 394 KPC (+) *K. pneumoniae* strains, 99.6% were inhibited by ceftazidime-avibactam, whereas only 61.9%, 59.6%, 58.4%, and 51.5% were inhibited by gentamicin, colistin, fosfomycin, and tigecycline, respectively. In addition, 19 (4.8%) of isolates exhibited a PDR phenotype and 124 (31.5%) exhibited an XDR phenotype [73].

The real-world efficacy of ceftazidime-avibactam in the treatment of KPC (+) mostly *K. pneumoniae* strains was shown in clinical post-marketing studies, proving that in general, when compared to the conventionally prescribed antibiotics, not only higher cure rates were observed, but also lower mortality rates [26,27,28,29,30,31,32]. A multicenter prospective observational study with 147 patients (140 with KPC-producing K. pneumoniae (KPC-Kp) and seven with OXA-48 K. pneumoniae isolates with a median MIC to ceftazidime-avibactam of 1 mg/L) was conducted between January 2018 and March 2019 in 14 tertiary hospitals located all over Greece. The APACHE II and SOFA scores at the onset of infection were 16.5 ± 7.6 and 6.7 ± 4.2, respectively, whereas 45 (30.6%) patients had an ultimately fatal, 21 (14.3%) patients had a rapidly fatal, and 81 (55.1%) patients had a non-fatal underlying disease. Half of the patients were hospitalized in the ICU (50.3%), 50 (34%) had septic shock and 97 (66%) sepsis (by Sepsis-3), highlighting the severity of infection burden. The outcome and mortality predictors were assessed in a variety of infections including mainly bacteremia (64.6%), cUTI (22.4%), HAP/VAP (25.2%), and cIAI (10.2%). The resistance rates reported were for meropenem, colistin, and tigecycline 99%, 34%, and 44%, accordingly; however, a PDR profile was not subjected in the analysis. Monotherapy was given to 68 (46.3%) patients whereas in 79 (53.7%) patients ceftazidime-avibactam was given in combination with at least another active in vitro antibiotic for a median duration of 13 days. At day 14, in 81% of patients clinical success was observed with microbiological eradication in 50.4% and presumed eradication in 37.4% with emergence of resistance in two patients (1.4%). Mortality rates at 14 and 28 days were 9% and 20%, respectively, the highest percentage observed being in pneumonia patients (38%). The study focused in particular on a subgroup of 71 patients with KPC-Kp BSI treated with ceftazidime-avibactam, which was matched by propensity score with an equal group of bacteremic patients treated with other than ceftazidime-avibactam antibiotics active in vitro. The 28-day mortality in the 71 patients treated with ceftazidime-avibactam versus that in the 71 matched patients given other active in vitro antibacterial was 18.5% vs. 40.8% (*p* = 0.005), respectively. As independent predictors of death, ultimately fatal disease, rapidly fatal disease, and Charlson comorbidity index ≥2 were determined, whereas therapy with CAZ-AVI was the only independent predictor of survival [31].

The largest study published in 2021 on the evaluation of ceftazidime-avibactam monotherapy was an Italian retrospective observational cohort comprised of 577 patients suffering mainly from bacteremia (*n* = 391, 67.7%), cUTIs (*n* = 71, 12.3%), lower respiratory tract infections (LRTI) (*n* = 59, 10.2%), and cIAI (*n* = 35, 6.1%) [32]. The Charlson comorbidity index ≥3 was observed in 85%, 24% were hospitalized in the ICU and 17.3% had septic shock. All were given ceftazidime-avibactam as monotherapy (*n* = 165) or with ≥1 other active in vitro antibiotic (*n* = 412), including fosfomycin (*n* = 92), tigecycline (*n* = 80), gentamicin (*n* = 68), meropenem (*n* = 69), colistin (*n* = 29), amikacin (*n* = 25), or other suitable antimicrobials (*n* = 18). All-cause mortality at 30 days post infection onset was 25%, without significant difference between the two groups (26.1% vs. 25.0%, *p* = 0.79). In multivariate analysis, the following factors being present at infection onset were positively connected with mortality: septic shock (*p* = 0.002), neutropenia (*p* < 0.001), INCREMENT score ≥8 (*p* = 0.01), lower respiratory tract infection (*p* = 0.04), and dose adjustment of ceftazidime-avibactam in case of renal insufficiency (*p* = 0.01). For the first time reported in the relevant literature, mortality was decreased whenever ceftazidime-avibactam was administered by prolonged infusion (≥3 h) in 246 patients (*p* = 0.006) as shown in 34.9% of the non-survivors vs. 45.2% of the survivors [32].

The administration of ceftazidime-avibactam in PDR *K.pneumoniae* infections is limited to case reports. Camargo. et al. [33] reported a case of BSI caused by PDR *K.pneumoniae* in an intestinal transplant patient. After failing multiple antimicrobial regimens (tigecycline, colistin, and meropenem in different combinations), the patient was successfully treated with a combination of ceftazidime-avibactam and ertapenem. In another case report, a combination of pre-adapted bacteriophage therapy with ceftazidime-avibactam was successful for a fracture-related infection due to pandrug-resistant *Klebsiella pneumoniae* [34]. The cure of recurring *K. pneumoniae* carbapenemase-producing PDR *Klebsiella pneumoniae* septic shock episodes due to complicated soft tissue infection using a ceftazidime-avibactam based regimen combined with meropenem, tigecycline, and gentamicin was successful in a case report [35]. Lastly, in a patient with severe pancreatitis, a carbapenem resistant PDR *K. pneumoniae* in the pancreatic tissue was identified and *bla*_KPC-2_ gene was detected. The patient was treated with a combination of ceftazidime-avibactam, metronidazole, and teicoplanin. The patient demonstrated clinical and microbiological response over the first 3 weeks; however, deteriorated after 6 weeks and died [36]. On the other hand, ceftazidime-avibactam has been administrated for PDR *K.pneumoniae* infections (BSI, UTI) in five neonates and children with a favorable outcome in all cases [37,38].

Resistance development to ceftazidime-avibactam is a great matter of concern. The worrisome phenomenon of ceftazidime-avibactam transferable resistance due to a novel VEB β-lactamase variant with a Lys234 Arg substitution in *K. pneumoniae* strains, five out of ten with a pan-drug resistant profile, has been published [74,75]. Epidemiological investigations revealed that the resistance was acquired independently from previous ceftazidime-avibactam exposure. Three patients developed an infection: two catheter-related bloodstream infections and one VAP. The salvage therapeutic regimen chosen was a combination of ceftazidime-avibactam with meropenem or aztreonam plus fosfomycin. The triple combination was successful in two of the cases, while the combination of ceftazidime-avibactam and meropenem was reported as a failure in the remaining one [75].

#### 2.2.2. Aztreonam-Avibactam

In the earliest in vitro evaluation, the new combined molecule was found very active against 114 *K. pneumoniae* MBL producing strains collected between 2016–2017 with an MIC of ≤2 mg/L [76]. In a more recent study, aztreonam-avibactam activity was tested against 8787 Enterobacterales collected consecutively in 2019 from 64 countries and 64 medical centers; 99.9% of strains were inhibited at ≤8 mg/L with 99.5% at ≤1 mg/L [77]. A still ongoing randomized phase 3 clinical trial in the evaluation of the efficacy and tolerability of aztreonam-avibactam in the therapy of serious infections due to MBL-producing Enterobacterales is expected to prove the real efficacy of the combination (clinical trial gov. identifier: NCT03580044). Currently and while awaiting AZ-AVI to be licensed, the combination of aztreonam and ceftazidime-avibactam has been given with very promising responses in patients with serious infections, in whom MBL producing bacteria were implicated. Dosages are depicted in Table 1. In the largest up-to-date study, which was prospective and observational, 102 cases with MBL bacteremia (82 with NDM and 20 with VIM) were included [59]. Results, when ceftazidime-avibactam plus aztreonam was given, were superior compared to active in vitro comparator antibiotics (mostly combination with colistin, tigecycline, fosfomycin, and aminoglycosides) with a lower 30-day mortality (19% vs. 44%, *p* = 0.01), as well as a lower number of clinical failures at day 14 [59]. In a case report, a PDR *K. pneumoniae* isolate encoding NDM-1, OXA-48, CTX-M-14b, SHV-28, and OXA-1 genes caused an infection of the cardiovascular implantable electronic device and right-sided infective endocarditis, that was treated successfully with the synergistic combinations of aztreonam with ceftazidime-avibactam for 6 weeks [60].

#### 2.2.3. Imipenem-Cilastatin-Relebactam

Against 137 strains of carbapenemase-producing Enterobacterales, relebactam reduced MICs of imipenem to 1 mg/L for 88% of the strains. Similarly, among 199 plasmids encoded KPC carbapenemases producing strains which were at 54% resistant to colistin, relebactam restored imipenem susceptibilities in 96.5% of isolates [78]. Regarding 295 KPC-Kp strains isolated in 2015–2016 from Greek hospitals, relebactam restored susceptibilities to 98% [79]. In the Restore-IMI-1 multicenter, a randomized, double-blind trial compared the safety and efficacy of imipenem-cilastatin-relebactam vs. colistin plus imipenem in 47 patients with imipenem-non-susceptible mostly cUTI and HAP/VAP infection. On day 28, a favorable clinical response was noticed in 71% vs. 40% with a 28-day mortality of 10% vs. 30%, respectively. To be pointed out, nephrotoxicity was observed in 10% vs. 56% (*p* = 0.002) [49]. No PDR infections treated with imipenem-cilastatin-relebactam has been reported to this date.

#### 2.2.4. Meropenem-Vaborbactam

In a phase III clinical trial (TANGO II), the efficacy and safety of meropenem-vaborbactam vs. the best available therapy (BAT) against CRE infections was evaluated in a randomized comparative study in which KPC-Kp represented 63.4% of resistant strains [80]. The cure rates of 65.6% vs. 33.3% (*p* = 0.03), with a 28-day all-cause mortality of 15.6% vs. 33.3% (*p* = 0.20) and microbiological cure reaching 65.6% vs. 40% (*p* = 0.09) were reported, respectively [80]. Accordingly, in two comparative prospective observational studies but with limited number of patients with CRE infections (20 and 40 patients, respectively), clinical success ranged from 65% to 70% with a 30-day mortality of 10% and 7.5% [44,45]. In a real-life based experience retrospective study with 131 patients, 105 were given ceftazidime-avibactam and 26 meropenem-vaborbactam, among whom 40% had bacteremia and the most common pathogen was KPC-Kp, and no significant differences either in clinical success or in mortality rates was reported [46].

## 3. Pandrug-Resistant *Acinetobacter baumannii*

### 3.1. Epidemiological Issues

Acinetobacter is an important cause of hospital-acquired infections, occurring mainly in ICU patients and among residents of long-term care facilities [81]. The most common infections encountered in the clinical setting are BSI, including catheter-relating bloodstream infections (CRBSI) and HAP, including VAP [82]. The most worrisome phenomenon of the last couple of years is the rise of PDR strains characterized as non-susceptible to all conventional antimicrobial agents [10]. In a systemic review of the current epidemiology and prognosis of PDR Gram—negative bacteria—a total of 526 PDR isolates were reported with 172 of them being PDR *A. baumannii*. The majority of PDR strains were isolated from ICU units, with a potential to cause hospital outbreaks, dissemination between hospitals and long-term facilities, as well as international transmission to other countries. PDR infections were associated with excess mortality, mounting up to 71%, and were independently high regardless of the infection source [9]. Notably, in a cohort study of 91 patients infected (*n* = 62) or colonized (*n* = 29) with PDR carbapenemase producing A. *baumannii* (CRAB), a three-fold increased hazard of mortality was observed in favor of patients with an infection caused by PDR CRAB [83]. Likewise, the comparison of patients with CRAB infections to patients with infections caused by carbapenem-susceptible *A. baumannii* was linked to increased mortality, prolongation of hospital stay, increased rate of ICU utilization, and hospital charges [5].

### 3.2. Therapeutic Options

#### 3.2.1. Antibiotics with Activity In Vitro against Carbapenemase Producing *A. baumannii*

The optimal therapeutic strategy for the management of carbapenemase producing *A. baumannii* (CRAB) infections exhibiting extensive drug-resistant phenotypes is very limited [84]. There is no “standard of care” treatment regimen for the therapy of CRAB. Sulbactam, meropenem, tigecycline, as well as polymyxins, the last-resort antibiotics in recent decades, have been used in critically ill patients for the treatment of CRAB infections [85]. Sulbactam, an irreversible β-lactamase inhibitor, has demonstrated activity against *A. baumannii* strains; unfortunately, it is administrated in combination with ampicillin (3 gr of ampicillin-sulbactam is comprised of 2 gr of ampicillin and 1 gr of sulbactam) [86]. For the treatment of CRAB infections, a dose of 9 gr ampicillin-sulbactam every 8 h with extended infusion of 4 h (total dose of 27 gr ampicillin-sulbactam in a patient with normal renal function) is suggested [85,87]. Polymyxins and mainly colistin is the most common antibiotic utilized in clinical practice for infections caused by CRAB [88,89,90]. In a systematic review and meta-analysis of polymyxins-based vs. non-polymyxins-based therapies in infections caused by CRAB, polymyxins-based therapies in terms of clinical efficacy had an advantage over non-polymyxins-based therapies (OR, 1.99; 95% CI, 1.31 to 3.03; *p* =0.001) [91]. The dosage of polymyxins is illustrated in detail in the International Consensus Guidelines for the Optimal Use of the Polymyxins [92]. Tigecycline, although it demonstrates being in vitro susceptible to *A. baumannii* [93], has been linked with higher mortality and lower microbiological eradication in two meta-analyses [94,95]. Improved clinical rates and lower mortality rates have been demonstrated when administrating a high dose of tigecycline (loading dose of 200 mg followed by 100 mg every 12 h) [96]. Thus, a high dose of tigecycline is recommended for the treatment of CRAB infections. Meropenem as a high-dose extended infusion of 3 gr every 8 h with a 3-h infusion has been utilized in combination therapy for the treatment of CRAB infections [85]. Lastly, in response to the medical need for new treatment options, cefiderocol and eravacycline, two new antimicrobial agents with in vitro susceptibility, have been recently approved [62,68]. The major problem is that the distribution of newly approved antimicrobial agents is suboptimal, with eravacycline being unavailable in Europe [97] and cefiderocol being used in compassionate access [98] or been recently launched in a minority of European markets (i.e., United Kingdom, Germany, and Italy) [99].

A respectable spectrum of antimicrobial combinations has been evaluated in vitro and in animal models, predominately based on polymyxins, rifampicin, fosfomycin, sulbactam, and carbapenems with promising results [100]. On the other hand, a variety of clinical studies evaluating in vitro synergy have failed to demonstrate superiority [101,102,103,104]. Indicatively, clinical studies comparing colistin monotherapy to colistin–rifampicin [101], colistin–fosfomycin [102], and colistin–meropenem combinations [103,104] depicted similar mortality rates with no significantly statistical difference in clinical cure. In a multicenter study from Italy, two hundred and ten ICU patients with infections due to XDR *A. baumannii* received either colistin methanesulphate (CMS) as monotherapy at a dose of 2 MU every 8 h intravenously, or CMS plus rifampicin 600 mg every 12 h intravenously. The thirty-day mortality in the combination and in the monotherapy arm was 43.3% and 42.9%, respectively, with no difference observed in terms of infection-related death and length of hospitalization [101]. In another study, ninety-four patients infected with CRAB (mostly HAP or VAP) were randomized to receive a combination of intravenous CMS at a dosage of 5 mg of colistin base activity/kg of body weight daily plus intravenous fosfomycin sodium at a dosage of 4 g every 12 h (47 patients in the combination group) or intravenous CMS (47 patients in the monotherapy group). Favorable clinical outcomes, mortality at the end of study treatment, and mortality at 28 days were not significantly different between groups [102]. The major drawback of both studies was the suboptimal dose of CMS (without a loading dose) utilized [101,102]. It is of great significance to analyze the two clinical trials evaluating the role of colistin monotherapy vs. colistin in combination with meropenem, due to large number of participants and the application of updated dose schemes [103,104]. The effectiveness of colistin monotherapy (9 million unit loading dose, followed by 4.5 million units every 12 h) to colistin–meropenem combination (2 gr prolonged infusion every 8 h) therapy for the treatment of severe infections caused by CRAB was evaluated in a randomized trial (with blinded outcome assessment). The majority of the patients had HAP, VAP, or bacteremia. Clinical failure rates for patients who received monotherapy versus combination therapy were 83% (125/151) vs. 81% (130/161) (*p* = 0.64), whereas mortality at 28 days was 46% (70/151) vs. 52% (84/161) (*p* = 0.4) for patients with *A. baumannii* infections [103]. In the second trial, 214 patients were enrolled in the colistin monotherapy arm and 211 in the meropenem-colistin combination arm. *A. baumannii* was the most common bacteria isolated (77%) and the most prevalent infections were nosocomial pneumonia and BSI. There were no differences between monotherapy and combination therapy in respect to 30-day mortality (43% vs. 37%, *p* = 0.21) and clinical failure rates (45% vs. 38%, *p* = 0.18) [104]. The results of both clinical trials strongly encourage the avoidance of colistin–carbapenem combination therapy for carbapenem-resistant *A baumannii* infections, regardless of the infection course.

#### 3.2.2. Salvage Treatment

A combination therapy with at least two agents, with in vitro activity whenever applicable, has been proposed by the IDSA guidelines for the treatment of moderate to severe CRAB infections [85]. The major issue, not referred to in the guidelines, is the treatment of PDR CRAB infections. Therapeutic options in these cases are based on in vitro and animal studies [100,105]. Two case series study with triple combination therapy have been reported for the treatment of PDR CRAB and are gradually implemented in clinical practice as salvage treatments due to the lack of other therapeutic choices [106,107], as shown in Table 2. The first study from Greece evaluated the triple combination therapy of intravenous high dose ampicillin-sulbactam (dose of 9 gr every 8 h), high dose of tigecycline (200 mg loading dose followed by 100 mg every 12 h), and intravenous CMS (9 million units loading dose, followed by 4.5 million units every 12 h) in 10 ICU patients with a VAP infection caused by *A. baumannii* with a PDR phenotype. The Charlson comorbidity index was ≥3 and the median APACHE score was of 23 ± 3. A successful clinical outcome was observed in 90% (9/10), whereas microbiological eradication was identified in 70% (7/10 patients). The 28-day mortality was of 10%, whereas nephrotoxicity was observed in one patient [106]. In another study, 20 patients with a median APACE score of 19.5 (range, 10–28) with infections caused by colistin-resistant *A. baumannii* were evaluated. The most common infections were VAP and bacteremia in 65% (13/20) and 10% (2/20), respectively. Three patients were characterized as colonization and were not treated, whereas the remaining 17 patients were treated in the majority with various CMS-based combination regimens. The most prevalent combination was a combination of carbapenem, ampicillin-sulbactam and CMS prescribed in seven patients. Mortality was depicted as lower in a statistical matter between triple combination and patients receiving other antimicrobial agents for the treatment of colistin-resistant *A. baumannii* (0% vs. 60%, *p* = 0.03) [108].

#### 3.2.3. New Antimicrobials

##### Cefiderocol

In the SIDERO-CR-2014-2016 surveillance in vitro study, European clinical isolates comprising MDR non-fermenter *A. baumannii* was tested against cefiderocol and 94.9% had a cefiderocol MIC ≤ 2 mg/L [109]. CREDIBLE-CR was a randomized, open-label, multicenter trial of cefiderocol (*n* = 101) and the best available treatment (BAT) (*n* = 49) for the treatment of severe infections (cUTI, nosocomial pneumonia, BSI, or sepsis) caused by carbapenem-resistant Gram-negative pathogens. In 118 patients in the carbapenem-resistant microbiological intent to treat (ITT) population, the most common baseline pathogen was *A. baumannii* in 46% (54/118). Cefiderocol was administrated as monotherapy in 83% (66/80) and combination therapy (mostly colistin-based regimens) was given in 71% (27/38) in the BAT arm. The clinical cure rates in the cefiderocol (22/49) and comparator (13/25) regarding *A. baumannii* were similar (45% vs. 52%). An increase in all-cause mortality was observed in patients treated with cefiderocol as compared to BAT. However, the greatest mortality imbalance disfavoring cefiderocol was noted in the nosocomial pneumonia subgroup, followed by BSI. The difference in 49-day mortality stratified for pathogen was the highest for *Acinetobacter* spp. (50% (21/ 42) vs. 18% (3/17) in cefiderocol and BAT-treated patients, respectively [110]. Deaths due to treatment failure in the cefiderocol group occurred more often in the patients infected with *Acinetobacter* spp. Of the 16 deaths due to treatment failure, 13 involved *Acinetobacter* spp. [109,110]. In conclusion, treatment failure was linked with infection caused by *Acinetobacter* spp., pulmonary infection at baseline, and by increases in cefiderocol MIC while on therapy [109,110]. An additional phase 3 trial, named APEKS-NP, evaluated hospital-acquired, ventilator-associated, or health-care-associated Gram-negative pneumonia and found cefiderocol was non-inferior to high-dose meropenem in patients. Fourteen-day all-cause mortality, clinical cure, and microbiologic eradication were similar between treatment groups for participants infected with *A. baumannii*; however, this group only comprised 16% of the study population, of which 66% of isolates were carbapenemase-resistant [111]. Cefiderocol has also been administrated as compassionate use in a limited number of case series with infections caused by XDR and PDR *A. baumannii* pathogens, resulting in a clinical success of 80% (20/25) [67,98]. Overall, the necessity of further studies to elucidate the true role of cefiderocol against *A. baumannii* infections in real life patients is needed.

##### Eravacycline

Eravacycline is a synthetic fluorocycline antibacterial agent that is structurally similar to tigecycline with two modifications at the D-ring of its tetracycline core [68]. In vitro activity of eravacycline against *A. baumannii* isolates (*n* = 2097) worldwide (from 2013 to 2017) revealed an MIC90s of 1 mg/L, demonstrating improved potency up to 4-fold greater than that of tigecycline [112]. Eravacycline has successfully completed clinical trial phase 3 for the treatment of cIAI; however, *A. baumannii* infections only comprised 3% of the total isolated pathogens [113]. Clinical studies with infections caused by CRAB reporting efficacy of eravacycline are lacking and are limited to one study. In a retrospective report of 93 adults hospitalized for pneumonia with DTR *A. baumannii*, 27 patients received eravacycline and were compared to those receiving the best available therapy. Eravacycline-based combination therapy had similar outcomes to the best available combination therapy. However, when taking under consideration patients with secondary bacteremia and coinfection with severe acute respiratory syndrome coronavirus-2 (SARS-CoV-2), eravacycline was associated with higher 30-day mortality (33% vs. 15%; *p* = 0.048), lower microbiologic cure (17% vs. 59%; *p* = 0.004), and longer durations of mechanical ventilation (10.5 vs. 6.5 days; *p* = 0.016), highlighting the avoidance of use in bacteremic patients [71]. However, eravacycline could be a suitable candidate for the treatment of cIAI caused by XDR, and even PDR pathogens. Therefore, further clinical studies addressing the efficacy of eravacycline in difficult-to-treat infections is required.

##### New β-Lactamase Inhibitor

Durlobactam, previously known as ETX2514, is a novel diazabicyclooctane class of β-lactamase inhibitor specifically designed to inhibit class D β-lactamases, in addition to class A and C enzymes. Durlobactam is combined with sulbactam, and targets infections caused by *A. baumannii* [21]. It has completed clinical trials in combination with sulbactam for the treatment of hospitalized adults with complicated urinary tract infection (cUTI) (Phase 2, clinicaltrials.gov identifier: NCT03445195) [114] and for the treatment of HAP and VAP caused by *A. baumannii* vs. colistin plus imipenem and the results are pending (Phase 3, clinicaltrials.gov identifier: NCT03894046).

## 4. *Pseudomonas aeruginosa* with Difficult-to-Treat Resistance

### 4.1. Epidemiological Issues

*Pseudomonas aeruginosa* is categorized among the ESKAPE pathogens and is considered one of the major causes of nosocomial infections caused by multi-resistant pathogens worldwide [115]. Resistance to last-resort colistin is still quite low. In vitro activity of colistin against isolates of *P. aeruginosa* collected in Europe as part of the INFORM global surveillance program from 2012 to 2015 revealed resistance to colistin < 0.5% [116]. Higher resistance rates have been observed in Greek isolates and are reported to be around 5–6% [117,118]. From the MagicBullet clinical study (2012–2015), fifty-three *P. aeruginosa* isolates from patients with HAP from 12 hospitals in Spain, Greece, and Italy were recovered. A minority was considered PDR (3.8%), whereas 19 (35.8%) were XDR and most of the isolates reported from Greece were PDR [118]. PDR strains of *P. aeruginosa* are extremely uncommon and are limited to 175 cases reported in a recent review [9]. Geographical distribution of PDR *P. aeruginosa* are mainly from Europe, Asia, and Australia, accumulating for 80, 52, and 34 cases, respectively. Almost one-third of the cases were defined in the ICU setting with a mortality rate ranging from 31–58% [9].

### 4.2. Therapeutic Options

There is a paucity of new classes of antibiotics active against *P. aeruginosa* resistant to carbapenems. Only four new antibiotics have a promising activity: ceftolozane-tazobactam, ceftazidime-avibactam, imipenem-cilastatin-relebactam, and cefiderocol [119]. However, most of those new antibiotics (excluding cefiderocol) are not active against MBL-producing *P. aeruginosa* isolates [120] and clinical experience with PDR *P. aeruginosa* is lacking. However, they are potent agents for the treatment of DTR *P. aeruginosa*.

#### 4.2.1. Ceftolozane-Tazobactam

MDR *P. aeruginosa* pathogens in the setting of phase 3 trials of ceftolozane-tazobactam treatment were 2.9% of uropathogens at baseline in cUTI, 8.9% in cIAI and in HAP, and VAP made up 25% of the study population [53,54,121]. In a multicenter, retrospective, cohort study at eight U.S. medical centers from 2015 to 2019, efficacy data of ceftolozane-tazobactam based on real-life experience was evaluated for the treatment of MDR and XDR *P. aeruginosa* isolates. Many patients had a high severity of illness at infection onset, with 50.6% residing in the ICU and a median APACHE II score of 21. The most common infection source was the respiratory tract in 62.9%. Clinical failure and 30-day mortality occurred in 85 (37.6%) and 39 (17.3%) patients, respectively [55]. A significant clinical experience of ceftolozane-tazobactam treatment exclusively in 101 various types of *P. aeruginosa* infections was reported from a retrospective study conducted in Italy (2016–2018). At the time of infection, 38.6% presented sepsis or septic shock and 23.8% were admitted to the ICU, with 56.4% classified as life-threating infections. Regarding *P. aeruginosa* strains, 50.5% were XDR and 78.2% were resistant to at least one carbapenem. An overall clinical success of 83.2% was depicted; however, lower rates were observed in patients with sepsis or undergoing continuous renal replacement therapy [56]. In a recent multicenter retrospective cohort of 95 critically ill ICU patients affected by severe infections due to *P. aeruginosa* (mostly nosocomial pneumonia) with different resistance patterns and 83.3% carbapenem-resistant (XDR 48.4% and MDR 36.8%), a favorable clinical response was observed in 71.6% of patients, with a microbiological eradication rate of 42.1% [57]. Therefore, IDSA guidance on the treatment of *P. aeruginosa* with difficult-to-treat resistance suggests ceftolozane-tazobactam therapy for cystitis, pyelonephritis, or cUTI, as well as for infections outside of the urinary tract [25], and the ESCMID guidelines on Gram-negatives recommend the use of ceftolozane-tazobactam in DTR *P. aeruginosa* infections with the obligation of in vitro susceptibility [122].

#### 4.2.2. Ceftazidime-Avibactam

In clinical trials with hospitalized patients with cUTI, cIAI, and HAP/VAP caused by *P. aeruginosa*, ceftazidime-avibactam was generally effective in terms of clinical cure and favorable microbiological response rates. In a pooled analysis of outcomes for patients with MDR Gram-negative isolates from the adult phase 3 clinical trials, ceftazidime-avibactam demonstrated similar efficacy to comparators against MDR *P. aeruginosa* [39]. The largest real-world study highlighting the clinical effectiveness of ceftazidime-avibactam in infections caused by MDR *Pseudomonas* spp. comprises 63 patients with *Pseudomonas* spp. infection. The most common infection source was the respiratory tract (60.3%). Clinical failure, 30-day mortality, and 30-day recurrence in terms of infections caused by *P. aeruginosa* occurred in 19 (30.2%), 11 (17.5%), and 4 (6.3%) patients, respectively [29]. The effectiveness of ceftazidime-avibactam for the treatment of 61 infections due to MDR/XDR *P. aeruginosa* was evaluated in a retrospective study. The median Charlson comorbidity index was 7, and 9.8% episodes were diagnosed in the ICU. The most common infection was lower respiratory tract infection (34.4%) and almost 15% were BSI and 50.8% presented with sepsis at symptom onset. Global clinical cure was achieved in 56 of 61 episodes (91.8%) and microbiological cure was achieved in 82.5% (33/40) of evaluable episodes, whereas mortality by day 30 was 13.1% [40]. In a systemic literature review with 150 cases of MDR/XDR or DTR *P. aeruginosa* infections treated with ceftazidime-avibactam, a favorable outcome ranging from 45–100% was depicted and superiority in a statistical manner vs. comparators was also illustrated [41]. Recent IDSA treatment guidelines for Gram-negative bacterial antimicrobial-resistant infections suggest ceftazidime-avibactam therapy in the settings of all DTR *P. aeruginosa* infections with limited therapeutic options [25]. However, the true efficacy of ceftazidime-avibactam against PDR *P. aeruginosa* is still lacking, due to deficit of reported cases.

#### 4.2.3. Imipenem-Cilastatin-Relebactam

In RESTORE-IMI 1 a phase 3, multicenter, double-blind trial, *P. aeruginosa* was the most common pathogen and was reported in 77% of cases with the majority of pathogens producing ESBL or *Pseudomonas*-derived cephalosporinases. Favorable overall response in terms of Pseudomonas infections was observed in 81% imipenem-cilastatin-relebactam and 62% colistin and imipenem patients (90% CI for difference, −19.8, 38.2), day 28 favorable clinical response in 71% and 40% (90% CI, 1.3, 51.5), and 28-day mortality in 10% and 30% (90% CI, −46.4, 6.7), respectively [49]. In a real-life retrospective, observational case series of 21 hospitalized patients treated with imipenem-cilastatin-relebactam, was conducted in 2020–2021 in the USA. The median APACHE II score was 21.5 and most patients (76%) were admitted to the ICU. The most common infections were respiratory tract infections, including HAP and VAP (52%), whereas bacteremia occurred in 29% of patients. The most prevalent pathogen was *P. aeruginosa* (16/21, 76%). Clinical cure occurred in 13/21 (62%) of patients treated with imipenem-cilastatin-relebactam, whereas mortality occurred in 33% (7/21) of patients [50]. The IDSA guidance on the treatment of *P. aeruginosa* with difficult-to-treat resistance suggests imipenem-cilastatin-relebactam therapy for cystitis, pyelonephritis, or cUTI, as well as for infections outside of the urinary tract [25]. However, the elucidation of the true clinical efficacy of imipenem-cilastatin-relebactam, as well as ceftazidime-avibactam in the era of PDR profiles is to be clarified in real-life studies.

### 4.3. Newer Antimicrobials

#### Cefiderocol

A CREDIBLE-CR study was initiated to evaluate cefiderocol’s safety and efficacy in patients with carbapenem resistant Gram-negative infections. Regarding *P. aeruginosa* infections, twelve (15%) were initiated in the cefiderocol arm and 10 (26%) in the BAT arm. All-cause mortality regarding *P. aeruginosa* infections was 35% (6/17) vs. 17% (2/12) in the BAT arm. Data reported also depicted that cefiderocol had a greater all-cause mortality compared with BAT at day 14 (6.6% difference), day 28 (18.4% difference), and day 49 (20.4% difference) of treatment [109]. In another phase III trial, APEKS-NP, when filtering results for *P. aeruginosa* as the cultured organism, a total of 24 (17%) and 24 (16%) were included in the cefiderocol and meropenem arm, respectively. All-cause mortality at 14-day was similar for both groups [8% vs. 13%, −4.7 (−22.4 to 12.9)] and clinical cure was 16/24 (67%) vs. 17/24 (71%) (−4.2, −30.4 to 22.0), respectively [112]. In real life conditions, seventeen patients with MDR *P. aeruginosa* treated with cefiderocol have been reported. The most common infection was associated with VAP infections (41.2%), occurring in COVID-19 patients, with 88.2% of the patients admitted to the ICU. Clinical cure and microbiological cure rates were 70.6% and 76.5%, respectively [66].

### 4.4. Salvage Therapy

Salvage therapy for the treatment of pandrug *P. aeruginosa* has been proposed with amikacin monotherapy adapted to the MIC of the pathogen. Two patients with severe sepsis (secondary BSI due to IAI and HAP) due to pan-resistant *P. aeruginosa*, were successfully treated with a high daily dose of amikacin, given as monotherapy, and combined with continuous venovenous hemodiafiltration (CVVHDF). Both patients were cured with a high daily dose (25 to 50 mg/kg) of amikacin to obtain a peak/MIC ratio of at least 8 to 10 (MIC of both isolates was 16 mg/L). CVVHDF provided no deterioration in renal function after treatment. High dosage of aminoglycosides combined with CVVHDF may represent a valuable therapeutic option for infection due to PDR *P. aeruginosa*; however, the limited number (only two cases) treated with this unique therapeutic agent [123] should be taken into consideration. Salvage therapeutic options are illustrated in Table 2.

In conclusion, the new β-lactam-β-lactamase inhibitors, i.e., cefepime-taniborbactam and aztreonam-avibactam, seem to be promising agents active in vitro against carbapenem-resistant *P. aeruginosa*, including pathogens producing MBL [124,125]. The combination cefepime–taniborbactam is a potential alternative treatment option for PDR infections, particularly those caused by MBL-producing isolates [124]. However, the combination of aztreonam plus avibactam appears to be an encouraging option against MBL-producing bacteria, especially for Enterobacterales, but much less so for *P. aeruginosa* infections [125].

**Table 2 antibiotics-11-01009-t002:** Salvage therapeutic options for DTR and PDR Gram-negative pathogens.

Antibiotic	Spectrum of Activity	Mechanism of Action	Dosage (Normal Renal Function)	Comments on DTR and PDR
**Ampicillin-sulbactam***plus***Tigecycline***plus***Colistin** [106]	**Activity against**: PDR *A. baumannii* [106]	Based on in vitro synergistic combinations [106]	**Ampicillin-sulbactam**: 9 gr IV every 8 h, infused over 3 h *plus* **Tigecycline**: 100 mg IV every 12 h (Loading dose of 200 mg IV tigecycline) *plus* **CMS (colistin)**: 4.5 MU IV every 12 h (Loading dose 9 MU IV CMS) [106]	***A. baumannii:*** Clinical studies on PDR *A. baumannii* (7 cases) with favorable clinical outcome of 100% [106]
**Ampicillin-sulbactam***plus***Meropenem***plus***Colistin** [107]	**Activity against**: PDR *A. baumannii* [105,107]	Based on in vitro synergistic combinations [100,105]	**Ampicillin-sulbactam**: 9 gr IV every 8 h, infused over 3 h *plus* **Meropenem**: 2 gr IV every 8 h, infused over 3 h *plus* **CMS (colistin)**: 4.5 MU IV every 12 h (Loading dose 9 MU IV CMS) [107]	***A. baumannii:*** Clinical studies on PDR *A. baumannii* (10 cases) with favorable clinical outcome of 90% [107]
**High daly dose of amikacin in combination with CVVHDF** [123]	**Activity against**: PDR *P. aeruginosa* [123]	Increased exposure regimen adapted to MIC of the pathogen [123]	**Amikacin**: 25 to 50 mg/kg IV (to obtain a peak/MIC of at least 8 to 10) *plus* **CVVHDF** [123]	***P. aerugonisa:*** Clinical data limited to two cases of PDR *P. aeruginosa* with secondary bacteremia due to cIAI and HAP with favorable clinical outcome [123]
**Double carbapenem** [13]	**Activity against**: *K. pneumoniae* producing KPC and OXA-48 [13,14,15,16,17,18,19,20]	Ertapenem higher affinity with the carbapenemase enzyme, acting as a suicide inhibitor, thus allowing higher levels of the other carbapenems (meropenem or doripenem) to be active in the vicinity of the pathogen [13]	1 gr IV ertapenem every 24 h, infused over 1 h *plus* 2 gr meropenem every 8 h, infused over 3 h [13,14,15]	***K. pneumoniae***: Real life clinical studies on XDR and PDR *K. pneumoniae* with favorable clinical outcome of 65% [13,14,15,16,17,18,19,20]

cIAI, complicated intrabdominal infection; CMS, colistin methanesulfonate; CVVHDF, continuous venovenous hemodiafiltration; DTR, difficult to treat resistance; HAP, hospital acquired pneumonia; IV, intravenous; KPC, Klebsiella pneumoniae carbapenemase; MBL, metallo-β-lactamase; MIC, minimum inhibitory concentration; MU, million international units; PDR, pandrug-resistant; XDR, extensively drug-resistant.

## 5. Conclusions

PDR and DTR Gram-negative infections have increasingly been reported globally in recent years and are linked to high mortality rates. There is “no standard of care” treatment regimen for the therapy of PDR and DTR Gram-negative infections, and therapeutic options are extremely limited. Synergistic combinations (double and triple combinations) seem quite promising; however, data are restricted to case reports and case series. The introduction of novel antimicrobials and mainly β-lactam-β-lactamase inhibitor combinations, as well as cefiderocol and eravacycline, are of great potential. However, the efficacy of novel antimicrobial agents for the treatment of PDR and DTR Gram-negative infections is to be elucidated in real-life studies in the near future.

## Figures and Tables

**Table 1 antibiotics-11-01009-t001:** Current and potential therapeutic options for DTR and PDR Gram-negative pathogens.

Antibiotic	Mechanism of Action	Spectrum of Activity	Mechanism of Resistance	Clinical Development Program and Approved Indications	Dosage (Normal Renal Function)	Comments on DTR and PDR
**Ceftazidime-Avibactam** (2.5 g: ceftazidime 2 g, avibactam 500 mg) [7,21]	Avibactam is a non–β-lactam β-lactamase inhibitor that inactivates some β-lactamases and protects ceftazidime from degradation [7,21]	**Activity against**: *K. pneumoniae* and *P. aeruginosa* producing ESBL, KPC, AmpC and some class D enzymes (OXA-10, OXA-48). No active against MBL, Acinetobacter spp [7,21]	Amino acid substitutions, insertions or deletions in three loops, the Ω-loop, the Val240 loop and the Lys270 loop and membrane impermeability of porin mutations [22]	**Approval**: FDA in 2015 [23] and EMA in 2016 [24] **Approved indications**: **FDA**: cIAI and cUTI in adults and pediatric age groups over 3 months of age, HAP and VAP in adults [23] **EMA**: cIAI and cUTI, HAP and VAP in adults and pediatric age groups over 3 months of age. Treatment of adult patients with bacteremia that occurs in association with, or is suspected to be associated with, any of the infections listed above. Treatment of infections due to aerobic Gram-negative organisms in adults and pediatric patients aged 3 months and older with limited treatment options [24]	2.5 g IV every 8 h, infused over 2–3 h [25]	***K. pneumoniae***: Real life clinical studies on XDR *K. pneumoniae* with favorable clinical outcome around 80%. Superiority of ceftazidime-avibactam against comparators [26,27,28,29,30,31,32]. PDR cases limited to case reports [33,34,35,36,37,38]. ***P. aeruginosa***: Real life clinical studies on XDR and DTR *P. aeruginosa* with favorable clinical outcome ranging from 45–100%. Superiority of ceftazidime-avibactam against comparators [29,39,40,41]. No PDR cases reported.
**Meropenem-Vaborbactam** (2g: meropenem 1g, vaborbactam 1g) [7,21]	Vaborbactam is a non-suicidal, boronic acid β-lactamase inhibitor with no antibacterial activity, preventing β-lactamases, such as KPCs, from hydrolyzing meropenem [7,21]	**Activity against**: *K. pneumoniae* producing ESBL, KPC, AmpC. No active against OXA-48-like, or MBL. As active as meropenem alone against *P. aeruginosa* [7,21]	Porin mutations in OmpK36 and OmpK35 and increased expression rate of the AcrAB-Toec efflux system [22]	**Approval**: FDA in 2017 [42] and EMA in 2018 [43] **Approved indications**: **FDA**: cUTI in adults [42] **EMA**: cIAI and cUTI, HAP and VAP in adults. Treatment of adult patients with bacteremia that occurs in association with, or is suspected to be associated with, any of the infections listed above. Treatment of infections due to aerobic Gram-negative organisms in adults with limited treatment options [43]	4 g IV every 8 h, infused over 3 h [25]	***K. pneumoniae***: Real life clinical studies on XDR *K. pneumoniae* with favorable clinical outcome around 65–70% [44,45,46]. No PDR cases reported.
**Imipenem-Cilastatin-Relebactam** (1.25 g: imipenem 500 mg, cilastatin 500 mg, relebactam 250 mg) [7,21]	Relebactam is a novel β-lactamase inhibitor of class with no intrinsic antibacterial activity, protects imipenem from degradation by some β-lactamases and Pseudomonas-derived cephalosporinase [7,21]	**Activity against**: *K. pneumoniae* and *P. aeruginosa* producing ESBL, KPC, AmpC and porin mutations. Diminished inhibitor activity against OXA-48. No activity against MBL and *A. baumannii* [7,21]	Porin loss of OmpK35 and OmpK36 as well as hyperexpression of *bla*_KPC_ [22]	**Approval**: FDA in 2019 [47] and EMA in 2021 [48] **Approved indications**: **FDA**: HAP and VAP in adults cUTI and cIAI in adult patients who have limited or no alternative treatment options [47] **EMA**: HAP and VAP in adults. Treatment of adult patients with bacteremia that occurs in association with, or is suspected to be associated with HAP or VAP in adults. Treatment of infections due to aerobic Gram-negative organisms in adults with limited treatment options [48]	1.25 g IV every 6 h, infused over 30 minutes [25]	***K. pneumoniae***: Real life clinical studies on XDR *K. pneumoniae* are limited [49] ***P. aeruginosa***: Real life clinical studies on DTR *P. aeruginosa* with clinical cure of 62% [50] No PDR cases reported.
**Ceftolozane-Tazobactam** (1 g ceftolozane/0.5 g tazobactam) [51]	Ceftolozane inhibits cell-wall synthesis via binding of PBPs. Tazobactam is a β-lactam sulfone that inhibits most class A β-lactamases and some class C β-lactamases [51]	**Activity against**: *K. pneumoniae* producing ESBL and AmpC. Activity against *P. aeruginosa* No activity against carbapenemase producing bacteria [51]	Modification of intrinsic AmpC-related genes and horizontally acquired β-lactamases that hydrolyse ceftolozane and are not inhibited by tazobactam, as well as modification of PBPs [51,52]	**Approval**: FDA in 2014 [53] and EMA in 2015 [54] **Approved indications**: **FDA**: cUTI, cIAI, HAP and VAP in adults [53] **EMA**: cUTI, cIAI, HAP and VAP in adults [54]	1.5 g IV every 8 h, infused over 1 h HAP/VAP: 3 g IV every 8 h, infused over 3 h [25]	***K. pneumoniae***: No activity against carbapenemase producing *K. pneumoniae* [51] ***P. aeruginosa***: Real life clinical studies on DTR *P. aeruginosa* with clinical cure of 62–83% [55,56,57] No PDR cases reported.
**Aztreonam-Avibactam** (Administrated currently as a combination of ceftazidime-avibactam and aztreonam until the approval of aztreonam-avibactam) [7,21]	Aztreonam is a monobactam combined with a novel non–β-lactam β-lactamase inhibitor. In contrast to most β-lactams, monobactams are not substrates for MBLs, whereas avibactam reversely inactivates most Class A and C and some D β-lactamase enzymes [7,21]	**Activity against**: *K. pneumoniae* producing ESBL, KPC, AmpC, OXA-48 and MBL. As active as aztreonam alone against *P. aeruginosa* and *A. baumannii*, including MBL-producing strains [7,21]	The production of β-lactamases (mostly AmpC variants in combination with NDM) and target modifications of PBP-3 [58]	**Phase 3**	**Ceftazidime-avibactam**: 2.5 g IV every 8 h, infused over 3 h *plus* **Aztreonam**: 2 g IV every 8 h, infused over 3 h (infused together) [25]	***K. pneumoniae***: Real life clinical studies on XDR *K. pneumoniae* (MBL producers) with lower 30-day mortality against in vitro comparator antibiotics and lower clinical failures [59] PDR cases limited to case report [60]
**Cefiderocol** **(1 g)** [7,61,62]	A new siderophore cephalosporin characterized as the “Trojan horse” because it creates a complex with the extracellular free ferric iron, leading to transportation of the drug through the outer cell membrane as a siderophore into the cell [7,61,62]	**Activity against**: *K. pneumoniae* producing ESBL, KPC, AmpC, OXA-48 and MBL. Activity against carbapenemase producing *P. aeruginosa* and *A. baumannii* [7,61,62]	The production of β-lactamases (mostly NDM, KPC and AmpC variants), porin mutations, mutations affecting siderophore receptors, efflux pumps and target modifications of PBP-3 [63]	**Approval**: FDA in 2019 [64] and EMA in 2020 [65] **Approved indications**: **FDA**: cUTI, HAP and VAP in adults [64] **EMA**: Treatment of infections due to aerobic Gram-negative organisms in adults with limited treatment options [65]	2 g IV every 8 h, infused over 3 h [25]	***K. pneumoniae***: No PDR cases reported. ***P. aeruginosa***: Real life clinical studies on DTR *P. aeruginosa* with favorable clinical outcome of 70.6% [66] No PDR cases reported. ***A. baumannii***: Real life clinical studies on XDR and PDR *A. baumannii* with favorable clinical outcome of 80% [67]
**Eravacycline** **(50 mg)** [7,68]	Eravacycline disrupts bacterial protein synthesis by binding to the 30S ribosomal subunit [68]	**Activity against**: *K. pneumoniae* producing ESBL, KPC, AmpC, OXA-48 and MBL. Activity against carbapenemase producing *A. baumannii.* No activity against *P. aeruginosa* [7,68]	The acquisition of genes encoding efflux pumps and the presence of ribosomal protection proteins, as well as target-site modifications such as the 16S RNA or certain 30S ribosomal proteins [68]	**Approval**: FDA in 2018 [69] and EMA in 2018 [70] **Approved indications**: **FDA**: cIAI in adults [69] **EMA**: cIAI in adults [70]	1 mg/kg/dose IV every 12 h [25]	***K. pneumoniae***: No PDR cases reported. ***A. baumannii***: Clinical studies on DTR *A. baumannii* with similar clinical cure rates compared to best available treatment. Higher mortality in bacteremic patients treated with eravacycline [71]

cIAI, complicated intrabdominal infections; cUTI, complicated urinary tract infections; DTR, difficult to treat resistance; EMA, European Medicines Agency; ESBL, extended-spectrum beta-lactamases; FDA, U.S. Food and Drug Administration; HAP, hospital acquired pneumonia; IV, intravenous; KPC, Klebsiella pneumoniae carbapenemase; MBL, metallo-β-lactamase; NDM, New Delhi metallo-β-lactamase; OXA, oxacillinase; PBP, penicillin-binding proteins; PDR, pandrug-resistant; VAP, ventilator associated pneumonia; XDR, extensively drug-resistant.

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
