# Peer review of "Current and Potential Therapeutic Options for Infections Caused by Difficult-to-Treat and Pandrug Resistant Gram-Negative Bacteria in Critically Ill Patients"

_antibiotics, 2022, doi:10.3390/antibiotics11081009_

Round 1

Reviewer 1 Report

I think the authors did a very good job of collating information from different sources in one document. They also indicated antibiotics to be used for pandrug resistance. On the first paragraph, I suggest that they write the number in words where the sentence start with a number and the yellow highlighted sentence that does not make sense or is incomplete.

Author Response

I think the authors did a very good job of collating information from different sources in one document. They also indicated antibiotics to be used for pandrug resistance.

 On the first paragraph, I suggest that they write the number in words where the sentence start with a number and the yellow highlighted sentence that does not make sense or is incomplete.

Response: The sentence was modified as for the number not be in the beginning of the sentence and the paragraph has been modified to accommodate to reviewer suggestion.

Reviewer 2 Report

Dear Authors and Editor,

I have read with interest the review “Therapeutic Option for Infections Caused by Pandrug Resistant Gram-negative in the ICU” by Giamarellou and Karaiskos, proposed for submission to Antibiotics.

In this review the authors revised the literature about infections due to drug-resistant Gram-negatives and their therapeutic options.

Despite the title, the paper is not throughout focused on the specific population of critically ill patients, but it seems rather a review of treatment options available for infections with difficult-to-treat organisms in general. I suggest narrowing the focus of the review to assess only studies dealing with patients admitted to the ICU or to clearly state which proportion of patients in any of the studies cited was critically ill.

General considerations:

- I would suggest extensive review of paragraphs formatting in order to improve readiness and logical congruence of the different topics treated. Moreover, I would suggest balancing the information provided for each drug (e.g. spectrum of activity, mechanisms of resistance, registration trials, real-life experiences) since the information are provided in a rather scattered manner in the different paragraphs about different pathogens.

-  I would suggest summarizing the treatment options evaluated by the authors in 1 or 2 tables, with references to the studies analyzed, again maintaining the focus only on the proportion of patients admitted to the ICU in each study.

Introduction:

- I suggest adding the definition of difficult-to treat (DTR) pathogens in the list of definitions of degrees of resistance.

- It is not clear whether the keyword used for literature search were used combined or one by one. Moreover, no keyword filter about critically ill/ICU patients was applied. I think authors should better state the methods used for literature search and results of such search (e.g. title/abstract screening vs full-text screening of papers).

- Section 2.1:  I would suggest changing the title of the section from “Carbapenem-resistant Enterobacterales” to “Carbapenem-resistant Klebsiella pneumoniae” since only this pathogen is discussed in the section. The introduction about the epidemiology of carbapenem-resistant K. pneumoniae and its mechanism of resistance is not easily readable (e.g. I would first describe Ambler’s classes of carbapenemases and later discuss their prevalence and distribution). How many of the reported cases of PDR K. pneumoniae infection involved critically ill patients admitted to the ICU? Do mortality data refer to all kinds of patients or only to critically ill ones? I think these data should be clearly stated as the focus of the review is particularly on patients admitted to the ICU.

- Section 2.2: Again, I would suggest changing the title of the section citing only K. pneumoniae and not Enterobacterales in general since only this pathogen is discussed in the section.

- Section 2.2.1: I would consider rephrasing the concept of salvage therapy expressed in the title and in the text. Most of the drugs cited are active against strains of (not exclusively PDR) carbapenem-resistant Enterobacterales, but express an unfavorable profile mainly in terms of toxicity and thus represent not preferred therapeutical options for the treatment of infections, especially among critically ill patients who frequently experience multiple toxicities due to concomitant treatments or multiorgan failure. The description of the study results in REF. 21 is not clearly stated (e.g. what kind of patients were included in the study? Which pathogens? Which resistance mechanisms/resistance profiles?). I would suggest focusing better on the results of studies dealing with PDR organisms (coherent with review title) and shorten the more “general” considerations about selectively carbapenem-resistant pathogens.

- Section 2.2.2: Same general considerations as of the above-mentioned point apply – better description of patients included and relevance with the ICU setting should be provided.

- Section 2.2.3: I suggest revising the sentence in line 167 addressing aztreonam as an inhibitor of MBL enzymes, which I find inaccurate. Moreover, I would suggest to revise the consequentiality of the description of the different b-lactamase inhibitors and their spectrum of activity.

- Section 2.3.1: Many data in the introduction of this section are redundant with data already reported in above sections. I would suggest removing duplicate information and significantly shorten the introduction to avibactam and its activity against KPC (since the focus of the review should be on PDR organisms). Again, data reviewed are always interesting but rather off-topic with regards to the title of the review.

- Section 2.4.2: I would suggest moving to this section some general data about spectrum of activity and phase 3 trials about eravacycline reported in section 3.2.3.2.

- Section 3.2.2: I find this section the most coherent with the title of the review, and encourage the authors to follow the format of this paragraph to review also the others paragraphs.

- Section 3.2.3.1: I would suggest removing duplicate information already provided in section 2.4.1 and focus only on available evidence on critically ill patients with A. baumannii infections.

- Section 4.1: As suggested above, I would recommend focusing this section on the epidemiology of P. aeruginosa infections in the ICU, with particular reference to the strains expressing PDR phenotype.

- Section 4.2: I find information provided in this paragraph rather scarce in comparison to all the other paragraphs, despite many more data about the treatment of P. aeruginosa infections are available in literature in comparison for example to A. baumannii. Although I would suggest the authors to deal with each topic in a concise manner, I recommend a more uniform way of presenting data reviewed (See General considerations above).

  • Section 4.3: Although data presented are of questionable clinical value, I again consider this paragraph as one of the most coherent with the title of the review. I encourage authors to adopt this format throughout the review, and in this specific section to add mention to the limitations of the soundness of observations of 2 case reports, adding mention to the site of origin of infection.

Overall, I would recommend the Editor to reject the work in its present form, but encourage the Authors to re-sumbit a new version of the work since the topic trated is indeed of great clinical interest and a revision of the available evidence to treat these difficult microorganisms is much appreciated.

Author Response

- I would suggest extensive review of paragraphs formatting in order to improve readiness and logical congruence of the different topics treated. Moreover, I would suggest balancing the information provided for each drug (e.g. spectrum of activity, mechanisms of resistance, registration trials, real-life experiences) since the information are provided in a rather scattered manner in the different paragraphs about different pathogens.

Response: Thank you for the comment. Information for each drug was modified to balance the data of each drug. However, when dividing pathogens, some aspects would be similar and information regarding the exact pathogen would be reported in a different way.

-  I would suggest summarizing the treatment options evaluated by the authors in 1 or 2 tables, with references to the studies analyzed, again maintaining the focus only on the proportion of patients admitted to the ICU in each study.

Response: A table was added to summarize the treatments.

Introduction:

- I suggest adding the definition of difficult-to treat (DTR) pathogens in the list of definitions of degrees of resistance.

Response: A definition was added (reference 8)

- It is not clear whether the keyword used for literature search were used combined or one by one. Moreover, no keyword filter about critically ill/ICU patients was applied. I think authors should better state the methods used for literature search and results of such search (e.g. title/abstract screening vs full-text screening of papers).

Response: The methods were updated.

- Section 2.1:  I would suggest changing the title of the section from “Carbapenem-resistant Enterobacterales” to “Carbapenem-resistant Klebsiella pneumoniae” since only this pathogen is discussed in the section. The introduction about the epidemiology of carbapenem-resistant K. pneumoniae and its mechanism of resistance is not easily readable (e.g. I would first describe Ambler’s classes of carbapenemases and later discuss their prevalence and distribution). How many of the reported cases of PDR K. pneumoniae infection involved critically ill patients admitted to the ICU? Do mortality data refer to all kinds of patients or only to critically ill ones? I think these data should be clearly stated as the focus of the review is particularly on patients admitted to the ICU.

Response: The title of the section was modified. The chapter was also modified and information on PDR were included.

- Section 2.2: Again, I would suggest changing the title of the section citing only K. pneumoniae and not Enterobacterales in general since only this pathogen is discussed in the section.

Response: The title of the section was modified

- Section 2.2.1: I would consider rephrasing the concept of salvage therapy expressed in the title and in the text. Most of the drugs cited are active against strains of (not exclusively PDR) carbapenem-resistant Enterobacterales, but express an unfavorable profile mainly in terms of toxicity and thus represent not preferred therapeutical options for the treatment of infections, especially among critically ill patients who frequently experience multiple toxicities due to concomitant treatments or multiorgan failure. The description of the study results in REF. 21 is not clearly stated (e.g. what kind of patients were included in the study? Which pathogens? Which resistance mechanisms/resistance profiles?). I would suggest focusing better on the results of studies dealing with PDR organisms (coherent with review title) and shorten the more “general” considerations about selectively carbapenem-resistant pathogens.

Response: The concept of salvage treatment was totally rephrased and information regarding mechanisms of resistance (where applicable) were added.

- Section 2.2.2: Same general considerations as of the above-mentioned point apply – better description of patients included and relevance with the ICU setting should be provided.

Response: The concept was totally rephrased and information regarding ICU setting and critically ill were added.

- Section 2.2.3: I suggest revising the sentence in line 167 addressing aztreonam as an inhibitor of MBL enzymes, which I find inaccurate. Moreover, I would suggest to revise the consequentiality of the description of the different b-lactamase inhibitors and their spectrum of activity.

Response: This section was omitted.

- Section 2.3.1: Many data in the introduction of this section are redundant with data already reported in above sections. I would suggest removing duplicate information and significantly shorten the introduction to avibactam and its activity against KPC (since the focus of the review should be on PDR organisms). Again, data reviewed are always interesting but rather off-topic with regards to the title of the review.

Response: Many data not relevant were omitted.

- Section 2.4.2: I would suggest moving to this section some general data about spectrum of activity and phase 3 trials about eravacycline reported in section 3.2.3.2.

Response: The data were moved to the above section to accommodate with reviewer suggestion.

- Section 3.2.2: I find this section the most coherent with the title of the review, and encourage the authors to follow the format of this paragraph to review also the others paragraphs.

- Section 3.2.3.1: I would suggest removing duplicate information already provided in section 2.4.1 and focus only on available evidence on critically ill patients with A. baumannii infections.

Response: Duplicate data were moved to the above section to accommodate with reviewer suggestion

- Section 4.1: As suggested above, I would recommend focusing this section on the epidemiology of P. aeruginosa infections in the ICU, with particular reference to the strains expressing PDR phenotype.

Response: Focus on ICU issues and PDR in this section was attempted. However, the data are extremely limited.

- Section 4.2: I find information provided in this paragraph rather scarce in comparison to all the other paragraphs, despite many more data about the treatment of P. aeruginosa infections are available in literature in comparison for example to A. baumannii. Although I would suggest the authors to deal with each topic in a concise manner, I recommend a more uniform way of presenting data reviewed (See General considerations above).

Response: Treatment of P.aeruginosa was updated.

Section 4.3: Although data presented are of questionable clinical value, I again consider this paragraph as one of the most coherent with the title of the review. I encourage authors to adopt this format throughout the review, and in this specific section to add mention to the limitations of the soundness of observations of 2 case reports, adding mention to the site of origin of infection.

Response: The missing data were added

Overall, I would recommend the Editor to reject the work in its present form, but encourage the Authors to re-sumbit a new version of the work since the topic trated is indeed of great clinical interest and a revision of the available evidence to treat these difficult microorganisms is much appreciated.

Reviewer 3 Report

I believe this is a well-organized paper that covers extensive treatment options available for pandrug-resistant Gram-negative bacteria. It is a good-quality review article with references to even the very recent publications.

I would like to make only few comments as follows.  

I recommend adding the term “isolates” or “bacteria” behind the phrase “Pandrug resistant Gram-negative” in the title.

I suggest adding a table that summarizes the overall CRE treatment options available. It would be very helpful for clinicians if there are information about the target patients, indications, and antibiotic options available for each resistant gene.

I suggest making a table to organize the approved indications and susceptible strains for the new antimicrobials.

Author Response

I believe this is a well-organized paper that covers extensive treatment options available for pandrug-resistant Gram-negative bacteria. It is a good-quality review article with references to even the very recent publications.

I would like to make only few comments as follows.  

I recommend adding the term “isolates” or “bacteria” behind the phrase “Pandrug resistant Gram-negative” in the title.

Response: the title was modified

I suggest adding a table that summarizes the overall CRE treatment options available. It would be very helpful for clinicians if there are information about the target patients, indications, and antibiotic options available for each resistant gene.

I suggest making a table to organize the approved indications and susceptible strains for the new antimicrobials.

Response: A table was added to summarize treatments

Reviewer 4 Report

Dears authors 

Carbapenem-resistant Gram-negative pathogens, with the main representatives Klebsiella pneumoniae, Acinetobacter baumannii and Pseudomonas aeruginosa are classified in the highest priority category for new treatments. The worrying phenomenon in recent years is the presence of pandrug resistant Gram-negative bacteria (PDR), characterized as not sensitive to all conventional antimicrobial agents. PDR Gram-negative infections are linked to high mortality and associated with nosocomial infections, mainly in the ICU. Treatment options for infections caused by PDR Gram-negative organisms are extremely limited and are based on clinical and serious cases. Here the current knowledge available on the treatment of PDR infections is discussed. A focus of the review focuses on rescue treatment and synergistic combinations (double and triple combinations), as well as increasing the MIC-adapted exposure regimen of the pathogen. The most available data on the new antimicrobials, including new combinations of β-lactam-β-lactamase inhibitors, cefiderocol and marvelacycline are presented comprehensively but the authors need to review a few things:
- Introduction: the contents and the drafting of the general part must be reformed to review the syntax of the topic.

- (lines 625-650) to deepen and understand the interaction between the presence and the mechanisms of drug resistance with those of formation and virulence of the biofilm is fundamental for addressing chronic bacterial infections and providing strategies for their management, adding these elements of literature suggested and discussing : PMID: 34572716 ; PMID: 34992394 ; PMID: 35169997 .

- Check the bibliographic entries throughout the text, some of which are non-compliant, review some entries in the references and necessarily insert those referred to in the previous point for the purpose of my acceptance.

- Review English grammar and in particular applied scientific English: in particular verb tenses and syntax in the discussion.

Author Response

 Introduction: the contents and the drafting of the general part must be reformed to review the syntax of the topic.

Response: Modification to the introduction were made.

  • (lines 625-650) to deepen and understand the interaction between the presence and the mechanisms of drug resistance with those of formation and virulence of the biofilm is fundamental for addressing chronic bacterial infections and providing strategies for their management, adding these elements of literature suggested and discussing : PMID: 34572716 ; PMID: 34992394 ; PMID: 35169997 .

Response: an Addition in section on epidemiology of Pseudomonas aeruginosa was conducted 

  • Check the bibliographic entries throughout the text, some of which are non-compliant, review some entries in the references and necessarily insert those referred to in the previous point for the purpose of my acceptance.

Response: References were updated

  • Review English grammar and in particular applied scientific English: in particular verb tenses and syntax in the discussion.

Response: Grammar and syntax were corrected

Round 2

Reviewer 2 Report

Dear Editor and Authors,

Thank you very much for providing a revised version of the review, now entitled “Current and potential therapeutic options for infections caused by difficult-to-treat and pandrug resistant Gram-negative bacteria in critically ill patients”.

I think that after revision the quality of the paper has much improved.

However, I still feel that lots of information might be omitted since they are quite off-topic from the subject of the review. Indeed, as clearly stated by the Authors in their replies, very little is known on the effective treatment of PDR/DTR infections. I think that, as little as it might be, this should be the only focus of the review, with enhanced synthesis on the paragraphs that serve as links between the different topics or as introductions.

Moreover, I would recommend in general shorter sentences to help readiness and clarity of the paper. I think that cutting down the length of the paper to at most 8-10 pages (references excluded) would help much. It might help to omit or sharply shorten the information in the text that are already pointed out in the table, eliminating duplicate information.

Please find below some more specific comments:

-        Section 2.2.1 (Salvage therapies): were the PDR isolates from the studies discussed susceptible to the salvage therapy administered (e.g. susceptible to colistin and/or tigecycline)? Were the three antimicrobials combined, found to be protective against mortality, in REF 21 all active against considered strains or were the strains at any rate resistant?

-        Acinetobacter section: rephrase lines 339-339 as part of the sentence is missing. Provide an extended definition of CRAB before first appearance (line 346). Please update information on the availability of cefiderocol in Europe, now licensed outside compassionate use in some countries (at least in Italy). While I think that cutting down most of the introduction would help, I would suggest developing in greater detail the paragraph on combination therapy, pointing out the different populations included, different susceptibility of pathogens and different outcomes as stated for the colistin-meropenem combination (lines 376-378). Please define in extent CMS before its first appearance in the text.

-        Table1: I would suggest re-editing the table, adding references inside the table text. I would suggest to summarize in the table many more data on the different drugs (see above) to shorten the text of the review. I would suggest the following columns, if feasible: antibiotic, mechanism of action, spectrum of activity, mechanism of resistance, approved indications, results of phase 3 trials. I would leave the comments on DTR and PDR pathogens to the text, where they can be fully addressed.

Overall, I would recommend acceptance after major revisions are provided.

Best regards,

Author Response

However, I still feel that lots of information might be omitted since they are quite off-topic from the subject of the review. Indeed, as clearly stated by the Authors in their replies, very little is known on the effective treatment of PDR/DTR infections. I think that, as little as it might be, this should be the only focus of the review, with enhanced synthesis on the paragraphs that serve as links between the different topics or as introductions. 

Moreover, I would recommend in general shorter sentences to help readiness and clarity of the paper. I think that cutting down the length of the paper to at most 8-10 pages (references excluded) would help much. It might help to omit or sharply shorten the information in the text that are already pointed out in the table, eliminating duplicate information.

Response: An effort to shorten the manuscript and to omit off-topic subjects was attempted, as well to minimize the manuscript and add as much information in revised table 1. A new table summarizing salvage treatments were also added

Please find below some more specific comments:

  •        Section 2.2.1 (Salvage therapies): were the PDR isolates from the studies discussed susceptible to the salvage therapy administered (e.g. susceptible to colistin and/or tigecycline)? Were the three antimicrobials combined, found to be protective against mortality, in REF 21 all active against considered strains or were the strains at any rate resistant?

Response: In both studies, isolates were defined as PDR, thus non-susceptible to all conventional antimicrobials (at the time of the study was conducted) and antimicrobial agents administrated were non-susceptible. In the first study, the authors defined empirical treatment, arbitrarily as “effective empirical therapy” the antimicrobial treatment administered before the microbiological documentation of the PDR infection that have led to clinical
improvement without need for treatment. A comment was added in the text. Regarding the other study, all isolated were defined PDR, therefore all resistant to intravenous agents administrated. 

  •        Acinetobacter section: rephrase lines 339-339 as part of the sentence is missing. Provide an extended definition of CRAB before first appearance (line 346). Please update information on the availability of cefiderocol in Europe, now licensed outside compassionate use in some countries (at least in Italy). While I think that cutting down most of the introduction would help, I would suggest developing in greater detail the paragraph on combination therapy, pointing out the different populations included, different susceptibility of pathogens and different outcomes as stated for the colistin-meropenem combination (lines 376-378). Please define in extent CMS before its first appearance in the text.

Response: Introduction was reduced to minimize the length of manuscript, therefore the suggested sentence was omitted. CRAB was defined as well as CMS. Cefiderocol availability was updated and was based on an authorized document published by Shionogi. Studies on combination treatment were analytically discussed. 

  •        Table1: I would suggest re-editing the table, adding references inside the table text. I would suggest to summarize in the table many more data on the different drugs (see above) to shorten the text of the review. I would suggest the following columns, if feasible: antibiotic, mechanism of action, spectrum of activity, mechanism of resistance, approved indications, results of phase 3 trials. I would leave the comments on DTR and PDR pathogens to the text, where they can be fully addressed.

Response: The table was updated with the majority of data to accommodate with reviewer suggestion. Also, a new table was introduced to summarize salvage treatments for PDR strains.

Round 3

Reviewer 2 Report

I feel that the quality of the paper has much improved. Please check the table for minor language/typing corrections.